# Arf-like Protein 2 (ARL2) Controls Microtubule Neogenesis during Early Postnatal Photoreceptor Development

**DOI:** 10.3390/cells12010147

**Published:** 2022-12-30

**Authors:** Cecilia D. Gerstner, Michelle Reed, Tiffanie M. Dahl, Guoxin Ying, Jeanne M. Frederick, Wolfgang Baehr

**Affiliations:** 1Department of Ophthalmology, University of Utah Health Science Center, Salt Lake City, UT 84132, USA; 2Department of Neurobiology & Anatomy, University of Utah, Salt Lake City, UT 84112, USA; 3Department of Biology, University of Utah, Salt Lake City, UT 84112, USA

**Keywords:** Arf-like protein 2 (ARL2), tubulin αβ-heterodimers, retina, rod photoreceptors, microtubule cytoskeleton (MTC), dynein heavy chain (DYNC1H1)

## Abstract

Arf-like protein 2 (ARL2) is a ubiquitously expressed small GTPase with multiple functions. In a cell culture, ARL2 participates with tubulin cofactor D (TBCD) in the neogenesis of tubulin αβ-heterodimers, the building blocks of microtubules. To evaluate this function in the retina, we conditionally deleted ARL2 in mouse retina at two distinct stages, either during the embryonic development (^ret^*Arl2*^−/−^) or after ciliogenesis specifically in rods (^rod^*Arl2*^−/−^). ^ret^*Arl2*^−/−^ retina sections displayed distorted nuclear layers and a disrupted microtubule cytoskeleton (MTC) as early as postnatal day 6 (P6). Rod and cone outer segments (OS) did not form. By contrast, the rod ARL2 knockouts were stable at postnatal day 35 and revealed normal ERG responses. Cytoplasmic dynein is reduced in ^ret^*Arl2*^−/−^ inner segments (IS), suggesting that dynein may be unstable in the absence of a normal MTC. We investigated the microtubular stability in the absence of either ARL2 (^ret^*ARL2*^−/−^) or DYNC1H1 (^ret^*Dync1h1*^−/−^), the dynein heavy chain, and found that both the ^ret^*Arl2*^−/−^ and ^ret^*Dync1h1*^−/−^ retinas exhibited reduced microtubules and nuclear layer distortion. The results suggest that ARL2 and dynein depend on each other to generate a functional MTC during the early photoreceptor development.

## 1. Introduction

The Arf-like (ARL) proteins of the Ras superfamily are small GTPases, discovered in *Drosophila* more than 30 years ago [1]. Soon thereafter, mammalian ARLs were cloned [2,3] and to date, >22 genes encoding ARL proteins have been identified in mammalian genomes [4]. ARL2 and ARL3 are the best characterized among ARLs (reviewed in [5]). ARL2 is expressed ubiquitously in yeast [6] and plants [7]. Gene mutations of the ARL2 ortholog, CIN4, in yeast cause aberrant chromosome numbers and the defects of a nuclear migration [6]. In *C. elegans*, the ARL2 ortholog, evl-20, regulates the MTC dynamics [8]. ARL2 is highly conserved throughout the eukaryotic evolution [9] and was shown to interact with PDE6δ [10,11,12,13], rootletin [14], BART (Binder of ARL2 or ARL2BP) [15,16,17,18], and CEP164 [19]. In humans, ARL2(R15L) is associated with MRCS syndrome (microcornea, rod-cone dystrophy, cataract, and staphyloma). The R15L mutation is located near the N-terminal helix and its inheritance is autosomal dominant [20]. The mechanisms associated with the disease are unknown. Compared to ARL2-WT, the binding affinity of ARL2(R15L) with ARL2BP decreased only by 18% (ref. [20]), suggesting that an interaction with ARL2BP is impaired.

Within the mammalian retina, ARL2 localizes in the photoreceptor IS, with an enrichment in the basal body area/rootlet [21]. ARL2 has a relatively weak affinity to nucleotides and presumably exchanges GDP with GTP without the assistance of the Guanine nucleotide Exchange Factor (GEF) [22,23]. The GAPs for ARL2 include ELMOD1-3 (ELMO Domain Containing 1–3), but these GAPs are not ARL2-specific [14,24,25,26]. In a cell culture, ARL2 associates with centrosomes throughout the cell cycle, and may be involved in the regulation of the mitochondrial fusion [27]. ELMOD2 and ARL2 localize to the photoreceptor rootlet associated with the centrosomal γ-tubulin ring complex (γ-TuRC) at the basal body [14] which is involved in the microtubule nucleation. The deletion of ELMOD2 in cell lines causes the delay of the recruitment of γ-TuRC and the microtubule nucleation from centrosomes [26]. The deletion of ELMOD3 in mice leads to a progressive hearing loss and abnormalities in the cochlear hair cell stereocilia [28,29].

ARL2 and the Tubulin Folding Cofactor D (TBCD) are key regulators in the assembly of αβ-tubulin heterodimers (see the Graphical Abstract), the building blocks of photoreceptor axonemes and the IS MTC [30,31,32,33]. The assembly of αβ tubulin heterodimers requires the participation of tubulin chaperones TBCA-D and the formation of a super-complex consisting of TBCC-E and ARL2-GTP [22,31,32,34]. The overexpression of TBCD caused the microtubule depolymerization that was inhibited by a co-expression with ARL2 [31,35]. The expression of ARL2 with a dominant activating mutation (ARL2-Q70L) caused the failure of the tubulin polymerization, with the loss of microtubules and the microtubule-based mitotic spindle, resulting in a cell cycle arrest in the M phase [33]. A transgenic model expressing ARL2-Q70L under a rod-specific promoter exhibited a reduced photoreceptor cell function and progressive rod degeneration in the third postnatal week [21].

In this article, we show that the embryonic deletion of ARL2 strongly affects the status of the photoreceptor MTC during the early development and distortion of the outer and inner nuclear layers. The microtubule’s formation was impaired at postnatal days 6 and 10, consistent with the important role of ARL2 in heterodimeric α- and β-tubulin biosynthesis. Outer segments (OS) did not form and electroretinography (ERG) a- and b-wave amplitudes were diminished. The cytoplasmic dynein levels of the IS were reduced or absent, suggesting that the dynein survival depends on the microtubules.

## 2. Materials and Methods

### 2.1. Mice

Egfp-*Cetn2* mice Stock No. 008234—CB6-Tg(CAG-EGFP/CETN2)3-4Jgg/J) and Six3Cre transgenic mice [36] (Stock No: 019755) were acquired from The Jackson Laboratory. iCre75 transgenic mice were generated at the University of Utah [37].

### 2.2. Generation of Retina- and Rod-Specific Knockout Mice

We acquired a cell line (UC-Davis KOMP repository, allele name Arl2tm1a(KOMP)Wtsi) with a gene trap flanked by loxP and FRT sites in intron 1 and a third loxP site in intron 3. Gene-trapped mice were generated at the University of Utah Transgenic Core Facility by a blastocyst injection. The germline transmission of the *Arl2*^GT^ allele was verified by PCR using primers upstream and downstream of the loxP site in intron 1. A floxed *Arl2* allele (*Arl2*^F/+^) was generated following the recombination of FRT-FLP with Flp-recombinase. Mice carrying the *Arl2*^F^ allele were mated with C57BL/6J mice to remove the *rd8* mutation inherent in KOMP mice [38]. *Arl2*^F/F^ mice were crossed with Six3Cre [36] or iCre75 transgenic mice [37] to generate retina-specific *Arl2* ^F/+^, Six3Cre (^ret^*Arl2*^+/−^) or rod-specific *Arl2* ^F/+^, iCre75 (^rod^*Arl2*^+/−^) heterozygous mice. The mice were then backcrossed to *Arl2^F^*^/F^ to generate experimental animals. For the identification of the centrioles and the connecting cilia, select mice were maintained on an Egfp-Cetn2 background.

### 2.3. Genotyping

The genomic DNA was extracted from the fresh tissue by dissolving the tail clips or whole retina from P8–12-day-old mice in 150 µL of tail lysis buffer at 50–60 °C for 1–2 h. The digests were centrifuged at 15,000 rpm for 5 min. A supernatant was added to an equal volume of isopropanol and centrifuged at 15,000 rpm for 5 min. The DNA pellet was rehydrated in 75 µL of H_2_O. The genotyping was achieved by a polymerase chain reaction with EconoTaq^®^ DNA polymerase (Lucigen, VWR International, Radnor, PA, USA) using the following primers. WT mice, CSD-F (5′- TGTCCTTCACTGGTTCCAAGTACCC), and CSD-ttr (5′- GACAAACTCATCACCCTTATGAAGCT) (483 bp); Post-Flip mice, CSD-F, and CSD-ttr (655 bp); Post-Flp and cre *Arl2* mice, CSD-F (5′-TGTCCTTCACTGGTTCCAAGTACCC), and CSD-R (5′-CTGCCCAGATAAATAAGAAGCCAAT) (741 bp); floxed Arl2 mice, CSD-loxF (5′-GAGATGGCGCAACGCAATTAATG), and CSD-R (341 bp); and the presence of Six3Cre and iCre75 transgenes: Six3CreFor (5′-TCGATGCAAGGAGTGATGAG) and Six3CreRev (5′-TTCGGCTATACGTAACAGGG) (550 bp). iCreFor (5′-GGATGCCACCTCTGATGAAG) and iCreRev (5′- CACACCATTCTTTCTGACCCG) (650 bp). An internal control was carried out with oIMR8744-fwd CAA ATG TTG CTT GTC TGG TG and oIMR8745-rev GTC AGT CGA GTG CAC AGT TT, generating an amplicon of the mouse T cell receptor delta chain gene. The internal positive controls show the presence of DNA in the negative PCR reaction (JAX protocol 22365: Standard PCR Assay—Tg(TcrAND)53Hed), Version 1.3).

### 2.4. Western Blot

The retinas were lysed in PHEM buffer (60 mM PIPES, 25 mM HEPES, 10 mM EGTA, and 4 mM MgSO4). A total of 30 ug of protein (Lowry assay) were separated by a 16% Bis-Tris gel using low-MW buffer (50 mM of MES (compound-2 *N*-morpholino) ethanesulfonic acid); 50 mM of Tris, 1 mM of EDTA, and 0.1% SDS). After the protein transfer to a 0.45 μm nitrocellulose membrane (ThermoFisher, Logan, UT, USA), the membrane was blocked using 5% milk in Tween Tris-buffered saline (TTBS) for 1 h and then incubated overnight at 4 °C in a polyclonal Arl2 antibody (diluted 1:1000, gift from Richard Kahn, Emory University) in the same buffer. The membrane was washed (10 min × 3) and incubated for 2 h in LI-COR secondary antibodies (mouse IR680 1:10 K and rabbit IR800 1:4 K). Following washes in 1X TTBS, the membrane was scanned using a LI-COR Odyssey imaging system (Lincoln NE).

### 2.5. Confocal Immunohistochemistry

The eyes were harvested on days P6, 10, and 15. The eyes were enucleated, the anterior segments were removed, and the eyecups immersion-fixed (4% paraformaldehyde in 0.1 M of phosphate buffer, pH 7.4) for 1 h on ice. The eyecups were cryoprotected by submersion in 15% sucrose in 0.1 M of phosphate buffer (30 min), followed by submersion in 30% sucrose in 0.1 M of phosphate buffer until the tissue equilibrated by sinking. The OCT-embedded eyecups were frozen and later cut at 14 μm. The sections were warmed at 37 °C for 30 min and washed (10 min × 3) in 1X TBS (Tris-buffered saline). The primary antibodies, raised in rabbit and directed against the following proteins, were used to label sections overnight at 4 °C in 5% goat serum−0.1% TritonX-100 in TBS: anti-ARL2 (gifts of Nick Cowan, NYU and Richard Kahn, Emory University); rhodopsin (Abclonal Technology, Woburn, MA, USA, 1:1000 dilution); DYNC1H1 (Proteintech Group Inc., Rosemont, IL, USA, 12345-1-AP, 1:250); anti-CTBP2 (RIBEYE, BD Biosciences 612044, Franklin Lakes, NJ, USA, 1:10,000); anti-GC1 (GUCY2E); and monoclonal antibody IS4 (Kris Palczewski, UC-Irvine). A visualization was achieved using either goat anti-rabbit or goat anti-mouse secondary antibodies tagged with Alexafluor 488, 555, or 647 (1:500) and incubated for 1 h at room temperature.

The tubulin immunohistochemistry was performed on tissue fixed, as described above, with the addition of ‘antigen retrieval’ after sectioning. Briefly, the slides with sections were warmed for 30 min and were then washed (10 min × 2) in TBS followed by an antigen retrieval for 5 min in 0.1% SDS in TBS at room temperature. The slides were then rinsed quickly (5 min) in TBS. The following antibodies were diluted in 5% goat serum in TBS and incubated overnight at 4 °C: rabbit anti-TUBA1A (Proteintech Group Inc., Rosemont, IL, USA, 14555-1-AP) 1:25, mouse anti-TUBB3 (Sigma-Aldrich, St. Louis, MO, USA, T8578) 1:100; monoclonal mouse anti-polyglutamylated tubulin IgM (Sigma-Aldrich, St. Louis, MO, USA, T9822) 1:250; and mouse anti-acetylated α-tubulin IgG2b (Sigma-Aldrich, St. Louis, MO, USA, T7451) 1:500.

Basal body labelling required a light fixation of 4% paraformaldehyde for 10 min only. The slides were incubated at 37 °C for 30 min, rehydrated in 1X TBS by washing (10 min × 3), blocked in 5% normal goat serum in 1X TBS, and incubated for either 1 h at RT or overnight at 4 °C. The primary antibodies were diluted in a blocking buffer (5% serum in TBS) to cover the wells and were incubated for 2 h at RT or overnight at 4 °C. The following antibodies were used: the CEP250/CNAP1 antibody (14498-1-AP, Proteintech Group Inc., Rosemont, IL, USA; rabbit anti-CEP164 (EMD Millipore, St. Louis, MO, USA, ABE 2621) 1:250; or anti-CEP164 (1:350, Sigma-Aldrich, St. Louis, MO, USA). After washing the slides in 1X TBS (10 min × 3), the secondary antibodies were diluted in a blocking buffer, applied to the sections, and the sections were incubated 1 h at room temperature in the dark.

Images were acquired using a Zeiss LSM 800 confocal microscope with a 63× objective. Some figures were post-processed with Airyscan. All of the genotypes of a given age and antibody were imaged at a single z-plane using identical settings for the laser intensity and the master gain. The digital gain = 1 for all of the images. The pinhole size was set for 1AU on the red channel (39 µm for the 40× objective). The post-processing of the non-saturated images consisted of equal adjustments to the brightness/contrast of the control and knockout images using Adobe Photoshop but without affecting the conclusions which were made. A red channel separation was obtained by isolating the R-channel in the “blender options” of Adobe Photoshop.

### 2.6. Electroretinography

Scotopic and photopic ERG measurements were performed using P15–P18 for Six3Cre experiments and P35 for the iCre75 experiments. Prior to ERG, the mice were dark-adapted overnight and anesthetized with an intraperitoneal injection of 1% ketamine/0.1% xylazine at a 10 µL/g body weight. The mice were kept warm during ERG by using a temperature-controlled stage. The scotopic and photopic responses were recorded as described [39,40,41] using a UTAS BigShot Ganzfeld system (LKC Technologies, Gaithersburg, MD, USA). The scotopic single-flash responses were recorded at stimulus intensities of −4.5 log cd s·m^−2^ [log candela seconds per square meter] to 2.4 log cd s·m^−2^). The mice were light-adapted under a background light of 1.48 log cd s·m^−2^ for 5 min prior to measuring the photopic responses. The photopic single-flash responses of the control and knockout were recorded at stimulus intensities of −0.1 log cd s·m^−2^ to 1.9 log cd s·m^−2^.

### 2.7. Statistical Analysis

We performed an unbalanced two-factor ANOVA to compare the experimental and control animals for their quantified A- and B-wave ERG measurements across multiple ages. A post hoc multiple comparison was performed using Tukey’s honestly significant difference criterion. The statistical significance was determined using an alpha value of *p* < 0.05. ERG statistics were computed using MATLAB’s statistical toolbox “anovan” and “multcompare” functions.

## 3. Results

### 3.1. Generation of the Retina-(^ret^Arl2^−/−^) and Rod-Specific (^rod^Arl2^−/−^) ARL2 Knockouts

Mouse ARL2 (184 residues) is a small GTPase featuring a G domain and small coiled-coil domains (Figure 1A) encoded by a ~6500 bp gene with 5 exons (Figure 1B). To enable conditional knockouts, we acquired a cell line in which a gene trap flanked by loxP and FRT sites was placed in intron 1 and a third loxP site was placed in intron 3 (Figure 1C). A floxed *Arl2* allele (*Arl2*^F^) (Figure 1D) is generated following the FRT-FLP recombination with Flp-recombinase. The deletion of ARL2 in the retina was achieved by mating with Six3Cre transgenic mice [36] expressing Cre recombinase at embryonic day 9 (E9) to yield *Arl2*^F/F^; Six3Cre knockouts (abbreviated as ^ret^*Arl2*^−/−^) (Figure 1E), or by mating with iCre75 transgenic mice expressing Cre recombinase under the control of the rhodopsin promoter to yield rod knockouts (^rod^*Arl2*^−/−^). In iCre75 mice, a Cre expression occurs during the second postnatal week when the photoreceptors are postmitotic. The deletion of exons 2 and 3 truncates ARL2 at residue 20 after exon 1 as exon 4 is out of frame. Genotyping (Figure 1F–H) confirmed the presence of loxP (F), the presence of Six3Cre (G, band of 450 bp), and the loss of exons 2 and 3 at P6 (H) in both of the knockouts (for details see Figure 1 legend and Methods). Western blots with the P10 ^ret^*Arl2*^−/−^ and P15 ^rod^*Arl2*^−/−^ retina confirmed the absence of ARL2 (Figure 1I,J).

### 3.2. ^ret^ARL2^−/− −/−^ Outer Nuclear Layers Display Abnormal Histogenesis

We generated plastic sections at P10-P15 to study the phenotype of the ^ret^*Arl2^−/−^* retinas relative to the controls. The control sections displayed normal photoreceptor IS and OS as well as correctly laminated nuclear layers (Figure 2A–C, left panels). By contrast, the nuclear layers, outer limiting membrane (OLM), and outer plexiform layers (OPL) of the ^ret^*Arl2*^−/−^ retinas were severely distorted. The ONL and INL thicknesses varied and the OPL was malformed with the absence of the outer segments, the OLM interruptions, and the photoreceptor nuclear migration into the subretinal space nuclei (Figure 2A–C, right panels). We observed a similar photoreceptor phenotype of retina ONL/INL distortion in P6 ^ret^*Dync1h1^−/−^* mice in which cytoplasmic dynein heavy chain 1 was deleted [41].

### 3.3. Cytoplasmic Dynein Is Unstable in ^ret^Arl2^−/−^ Photoreceptors

Based on the ^ret^*Arl2*^−/−^ phenotype with an ONL distortion (Figure 2), we suspected that the cytoplasmic dynein, a multi-subunit complex organized around the heavy chain DYNC1H1, may be affected by the distorted/damaged MTC. The dynein transports cargo toward the minus-end of the microtubules at the basal body and are essential for the retina lamination, nuclear positioning, vesicular trafficking, and inner/outer segment elaboration [41]. The immunohistochemistry (IHC) of P6 and P10 control retina cryosections with anti-DYNC1H1 showed the normal lamination of the nuclear layers (Figure 3A–D, left panels). A low magnification image of the entire P6 retina reveals the presence of DYNC1H1 in IS, OPL, and IPL of the control retina (Figure 3A, left panel), but distorted ONL/INL layers and the absence of DYNC1H1 (green) in the IS and OPL of the ^ret^*Arl2*^−/−^ retina (Figure 3A, right panel). At P6, the control OSs begins to form as guanylate cyclase 1 (GC1, gene nomenclature GUCY2E), a component of the phototransduction cascade, is detectable in budding OSs (Figure 3B, left panel. inset). GUCY2E is undetectable in ^ret^*Arl2*^−/−^ sections (Figure 3B, right panel). In the P6 controls, the MTC labeled with anti-Ac-α-tubulin is well developed (Figure 3C, left panel) but severely distorted in the absence of ARL2 (Figure 3C, right panel). At P10, the C-terminal binding protein 2 (CTBP2 alias RIBEYE) is located in the synaptic region of the controls (Figure 3D, left panel). The ^ret^*Arl2*^−/−^ (Figure 3D, right panel) results reveal a thinning or the absence of OPL, indicating defective synaptogenesis. Taken together, the ^ret^*Arl2*^−/−^ retina cryosections displayed distorted nuclear layers, the reduction in DYNC1H1 in the IS, and malformed microtubules.

### 3.4. ^ret^Arl2^−/−^ versus ^rod^Arl2^−/−^ Electroretinography

The functional analysis of the ^ret^*Arl2*^−/−^ retinas by the pan-retina ERG at P15 and P18 demonstrated that the average scotopic a- and photopic b-waves at 1.5 log cd s/m^2^ were severely diminished in the knockout (Figure 4A,B, red traces), consistent with the absence of OS in the central retina. The average scotopic a- and photopic b-wave amplitudes as a function of the light intensity (from −1.6 to 2.4 log cd s/m^2^) were nearly extinguished (Figure 4C,D). Residual scotopic and photopic a- and b-waves are attributed to the formation of the short photoreceptor OS, commonly observed with Six3Cre knockouts in the retina periphery where the expression of Six3Ce is delayed [41,42]. By contrast, the ^rod^*Arl2*^−/−^ average ERG traces at P35 (*n* = 5) and scotopic a- and photopic b-wave amplitudes as a function of the light intensity (Figure 4E–H) indicate the near-normal function of ^rod^*Arl2*^−/−^ rods and cones. Scotopic a-waves and photopic b-waves were indistinguishable at intensities from −4.5 to 2.4 log cd s m^−2^, ruling out significant morphological changes in the inner and outer segments. We conclude that the ^rod^*Arl2*^−/−^ rod and cone OSs developed normally despite the reduced αβ-tubulin heterodimer biosynthesis.

### 3.5. Effect of ARL2 Deletion on Pericentriolar Material

As ARL2 interacts with the basal body and the rootlet [14], we investigated the effects of the deletion of ARL2 on the localization of the basal body (BB) markers CEP164 and CEP250, employing EGFP-CETN2 to serve as a centriole and connecting cilium marker [39,41]. CEP164 is a distal appendage protein surrounding the BB distal end [39,43,44] (Figure 5A) and is required for BB docking during the development of the photoreceptor [14,45]. BB docking and CC extension in ^ret^*Arl2*^−/−^ photoreceptors is indistinguishable from the controls (Figure 5A′,B′) but the connecting cilia do not extend axonemes and OS. Rather than being located at the inner segment cortex, ^ret^*Arl2*^−/−^ BBs mislocalize within the ^ret^*Arl2*^−/−^ ONL (Figure 5B).

As ARL2 and rootletin are known interactants [14], we investigated whether ARL2 is involved in the centriole cohesion. The centrosome linker proteins CEP250 (C-Nap1), rootletin (CROCC1), and CEP68 connect the mother and daughter centrioles during ciliogenesis [46,47]. We probed P10 *Arl2*^F/F^; Egfp-Cetn2^+^ (Figure 5C) and ^ret^*Arl2*^−/−^; Egfp-Cetn2^+^ cryosections (Figure 5D) with anti-CEP250. The results show that C-NAP1 localizes between the mother and daughter centrioles of both the control and ^ret^*Arl2*^−/−^ photoreceptors (Figure 5C’,D’), suggesting that the absence of ARL2 does not affect centriole cohesion.

### 3.6. Microtubules Are in Disarray in the ARL2 Knockout

The results shown in Figure 3C are consistent with an MTC defect, generated by the absence of ARL2. The photoreceptor MTC is controlled by the microtubule organizing center (MTOC) consisting of the basal body and daughter centriole. The microtubule minus ends are anchored in the pericentriolar matrix and the basal body [48]. To test the status and stability of the ^ret^*Arl2*^−/−^ MTC in the retina ONL and INL, we used anti-acetylated α-tubulin (Ac-Tub) and anti-polyglutamylated tubulin (polyE-Tub) antibodies to probe the P6 and P10 retina cryosections (Figure 6). The acetylation of lysine 40 of α-tubulin and the addition of glutamate and glycine to both α- and β-tubulin, referred to as polyglutamylation and polyglycylation, are known to stabilize the microtubules (reviewed in [49]. Ac-Tub (white) and polyE-Tub (red) are labeled strongly at P6 and traverse the ONL of the *Arl2*^F/F^ retina (Figure 6A,C). At this age, Mueller glia, which penetrate the ONL at later ages, are not detectable in the ONL/INL [41] (see discussion). At P10, photoreceptor OS are emerging, but the labeling of the connecting cilia and axonemes by Ac-Tub and PolyE is relatively weak (Figure 6C, right panel, inset), in contrast to a previous report where polyglutamylated microtubules are enriched at the photoreceptor connecting cilium [50]. At P6 (Figure 6B), the ^ret^*Arl2*^−/−^ ONL is strongly distorted, acetylated, and the polyglutamylated microtubules appear disrupted and significantly reduced. At P10, ^ret^*Arl2*^−/−^ ONL appears to stabilize to a degree, but the ONLs thickness is reduced, the connecting cilia are not forming (Figure 6D), and the photoreceptors are degenerating. The microtubules are severely reduced at P6 (Figure 6B) and P10 (Figure 6D), consistent with the reduced levels of tubulin heterodimers in the absence of ARL2 [22,31,34]. The results suggest that the ^ret^*ARL2*^−/−^ heterodimer output and MT formation is decreased but not abolished.

### 3.7. Microtubule Cytoskeleton Is Unstable in Both ^ret^ARL2^−/−^ and ^ret^Dync1h1^−/−^ Retina

The results suggest that the absence of ARL2 early in the postnatal development lowers the tubulin heterodimer levels and affects the MTC maturation, which in turns affects the dynein function. We investigated whether the deletion of DYNC1H1, the key subunit of the cytoplasmic dynein, would affect the MTC similarly, as observed with the ablation of ARL2. We compared the status of the P10 MTC in ^ret^*ARL2*^−/−^ and P6 ^ret^*Dync1h1*^−/−^ photoreceptors compared to the controls by immunohistochemistry using anti-TUBA1A (α-1A tubulin, red) and anti-TUBB3 (β3-tubulin, green) antibodies (Figure 7). In the control *Arl2*^F/F^ and Dync1h1^F/F^ sections, α- and β-tubulin form a normal photoreceptor cytoskeleton (Figure 7A,C). In the ^ret^*Arl2*^−/−^ (Figure 7B) and ^ret^*Dync1h1*^−/−^ sections (Figure 7D), the MTC is disorganized and the microtubules are severely reduced. The ^ret^*Dync1h1*^−/−^ sections show a reduced ONL thickness and the near-complete absence of microtubules (Figure 7D). This result indicates that dynein and ARL2 are essential for the microtubule and MTC stabilization.

## 4. Discussion

To assess the status of the photoreceptor microtubules, we generated the conditional retina and rod knockouts deleting ARL2 before and after ciliogenesis (Figure 1). The absence of ARL2 during the early retina development affected the stability of ONL and INL (Figure 2 and Figure 3) and the MTC in the photoreceptors as early as P6 (Figure 6 and Figure 7). OSs did not form in the central retina and the ERG responses were diminished (Figure 4). The ONL/INL distortion also affected the formation of the OPL and synapses (Figure 3D). The function of the CEP164 (promoting the docking of the basal body to the apical membrane and generating the connecting cilium) and CEP250 (required for the centriole cohesion) were not affected, but the connecting cilia did not localize to the cortex of the cell and were found to be misplaced within the ONL (Figure 5).

Apart from its function as a release factor for prenylated proteins bound to PDE6D [51], ARL2 is involved in the biosynthesis of αβ-tubulin heterodimers, the building blocks of the microtubules [22,52,53]. The history of ARL2 as a factor in tubulin heterodimer biosynthesis is well documented [30,34,54,55]. The assembly of the αβ-tubulin dimer occurs in a folding cycle following the biosynthesis of α- and β-tubulins, a highly controlled process (see graphical abstract). The assembly requires the orchestrated action of a set of proteins of the chaperonin-containing TCP1 complex (CCT) [56], as well as tubulin-specific co-factors, i.e., TBCA-D [31,32,52]. The in vitro biochemical studies revealed that α-tubulin binds to TBCB and β-tubulin to TBCA that are replaced by TBCD and TBCE [32,52]. ARL2, when bound to TBCD, exchanges GDP with GTP and plays a critical role in the formation of a super complex consisting of TBCC/TBCE/α-tubulin and TBCD/β-tubulin/ARL2-GTP. Following the release of TBCD and TBCE, triggered by GTP hydrolysis, α-tubulin and β-tubulin are released as the heterodimers. The consequence Is that, in the absence of ARL2, the tubulin heterodimer levels are reduced, thereby affecting the microtubule’s assembly (see graphical abstract). Low levels of heterodimers during the early development of the photoreceptor effectively curtail the production of MT filaments which are necessary to stabilize dynein.

In Drosophila melanogaster neuroblasts, Arl2 physically associates with tubulin cofactors C, D, and E. An Arl2 RNA interference, Arl2-GDP expression, or Arl2 deletions caused microtubule abnormalities, suggesting that ARL2 regulates the microtubule’s growth [57]. Chen et al., conclude that Arl2 and its cofactors are probably responsible for exquisitely regulating the free tubulin heterodimer levels in the cell and the localization of dynein at the microtubules. However, the knockdown of ARL2 by siRNA in a tissue culture produced conflicting results. The deletion of ARL2 in immortal HeLa cells did not produce a phenotype [33], but silencing ARL2 in human neural progenitor cells provoked an apoptotic phenotype [58]. The depletion of Arl2 by siRNA in human cell lines resulted in a TBCD-mediated microtubule disruption [31]. ARL2 siRNA significantly reduced the cilia lengths of ARPE19 cells [15].

We observed that the cytoplasmic dynein levels in the photoreceptor IS were reduced in ^ret^*Arl2*^−/−^ photoreceptors compared to the controls (Figure 3), suggesting that cytoplasmic dynein may become unstable in the absence of microtubule tracks. We previously observed that the ablation of the dynein-heavy chain in the retina negatively affected the MTCs stability [41]. We therefore tested whether the microtubules and dynein were interdependent for their stability. Floxed *Dync1h1* P6 retina cryosections probed with anti-TUBB3 and anti-TUBA1A revealed a stable MTC comparable to the Arl2 controls (Figure 7C). By contrast, the *Dync1h1*^−/−^ sections revealed a severely attenuated ONL with an unstable MTC with few intact microtubules (Figure 7D). The effect on the microtubules in the ^ret^*Arl2*^−/−^ sections is less dramatic, presumably because the removal of ARL2 reduces but not obliterates the tubulin heterodimer levels. The MTC instability in the ONL could be in part caused by the degenerating Mueller glia. However, at P6, Mueller glia processes are not detectable in the ONL/INL and begin to penetrate the ONL after P10, as we have shown using anti-glutamine synthase as a marker [41]. Therefore, the instability of the ONL MTC is most likely caused by the reduced availability of tubulin heterodimers in photoreceptor axons. The direct interaction of ARL2 with cytoplasmic dynein has, to the best of our knowledge, not previously been reported. A link between the microtubule’s stability and dynein-2 (IFT dynein) has been provided in *Tetrahymena*. The microtubules of a microtubule-dynein-2 complex appear to be intact under conditions that result in a microtubule depolymerization. When dynein was dissociated from the complex with the addition of ATP, no microtubules were found in specimens under the same depolymerizing conditions, suggesting a microtubule catastrophe [59].

Surprisingly, the ^rod^*Arl2*^−/−^ retina (rod knockout) is stable and functional in the first six postnatal weeks (Figure 4E–H). The tubulin heterodimer biosynthesis is expected to slowly decrease in ^rod^*Arl2*^−/−^ mice in the first two postnatal weeks before the Cre expression under the control of the rhodopsin promoter begins. An MTC can be established, and the photoreceptors are mature. After the onset of the expression of Cre, this is visible by immunohistochemistry at around P16 [40]. ARL2 is knocked out but the tubulin heterodimer apparently is still produced at levels sufficient to maintain the MTC. These findings suggest that the effect of the inactivation of ARL2 is defined by a developmental switch, ending the rod maturation phase (P6-P16). In the early developmental phase, photoreceptors are building the MTC, requiring high levels of tubulin, establish nuclear layers, and IS and OS. In the mature phase (>P16), rods are more stable and can tolerate the loss of ARL2 and reductions in the heterodimers. An example for a developmental switch is the PKD1 (polycystin-1) knockout. In a mouse *Pkd1* conditional knockout model, the inactivation of PKD1 before postnatal day 13 results in the formation of severe cystic kidney cysts within 3 weeks, whereas an inactivation at day 14 and later results in cysts only after 5 months. These findings suggest that the effects of a *Pkd1* inactivation are defined by a developmental switch that signals the end of the terminal renal maturation process [60].

## 5. Conclusions

ARL2 is a multifunctional protein expressed in vertebrates, invertebrates, yeast, and plants. Our results suggest that during the early development of the photoreceptor, the photoreceptor microtubule cytoskeleton is unstable due to the insufficient production of tubulin heterodimers. We suggest that the ^ret^*Arl2*^−/−^ phenotype appears to be a consequence of a triple insult: an impaired tubulin heterodimer synthesis, reduced microtubule assembly, giving rise to a nonfunctional MTC, and the downregulation of dynein. By contrast, the ^rod^*Arl2*^−/−^ photoreceptors are stable and can tolerate the loss of ARL2 and reductions in the heterodimers.

## Figures and Tables

**Figure 1 cells-12-00147-f001:**
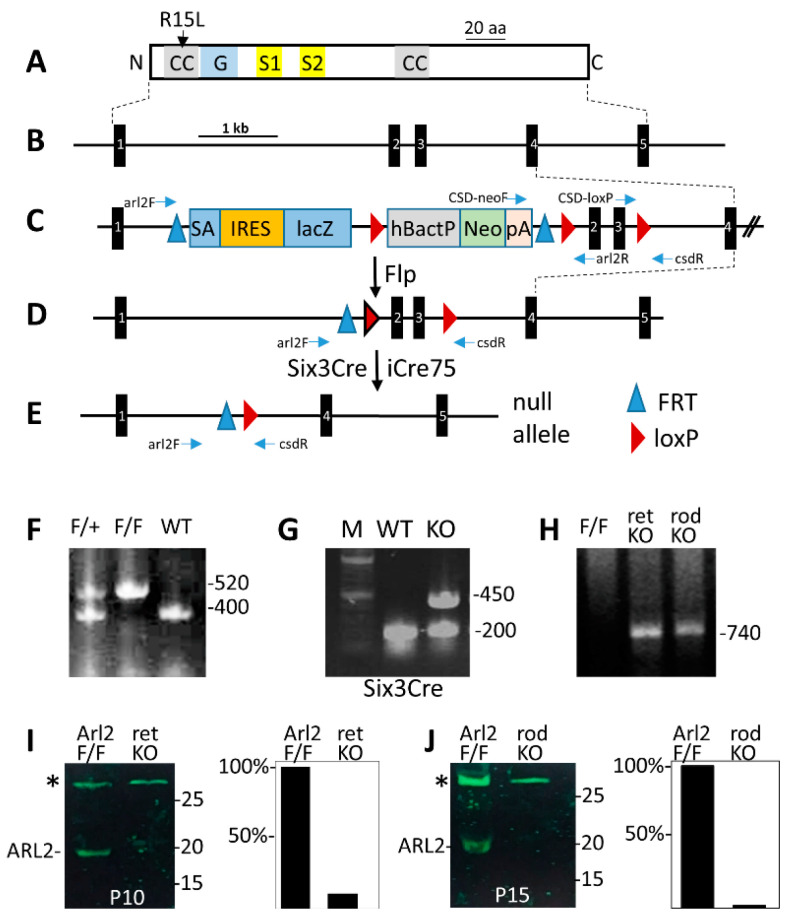
Generation of *Arl2* conditional knockouts. (**A**) schematic of mouse ARL2 protein and its functional domains. CC, coiled coil domain; G, guanine nucleotide binding domain; S1 and S2, switch 1 and switch 2; R15L, mutation linked to retina disease. (**B**) the mouse *Arl2* gene with 5 exons. (**C**) the gene trap is located in intron 1. The gene trap is flanked by FRT sites. LoxP sites are flanking the NEO cassette and a third loxP site is placed in intron 3. Horizontal blue arrows delineate approximate positions of genotyping primers. (**D**) floxed allele. (**E**) null allele. (**F**) genotyping floxed allele in tail DNA with arl2-F and arl2-R yielding a WT amplicon of 520 bp and a floxed amplicon of 400 bp. (**G**) presence of Six3Cre in tail DNA using Six3Cre–F and Six3Cre–R yielding an amplicon of 450 bp. The bands of 200 bp are an internal positive control (see Methods). (**H**) genotyping the Six3Cre (ret) and iCre65 (rod) knockout allele with CSDF and CSDR using retina DNA as a template. The diagnostic fragment of 740 bp is absent in Arl2^F/F^. (**I**,**J**) Western blots (left panels) with anti-ARL2 antibody using P10 ^ret^*Arl2*^−/−^ retina lysate (**I**) and P15 ^rod^*Arl2*^−/−^ retina lysate (**J**). Polypeptides marked with asterisks are nonspecific polypeptides serving as loading controls. (**I**,**J**) right panels are Image J density scans of ARL2.

**Figure 2 cells-12-00147-f002:**
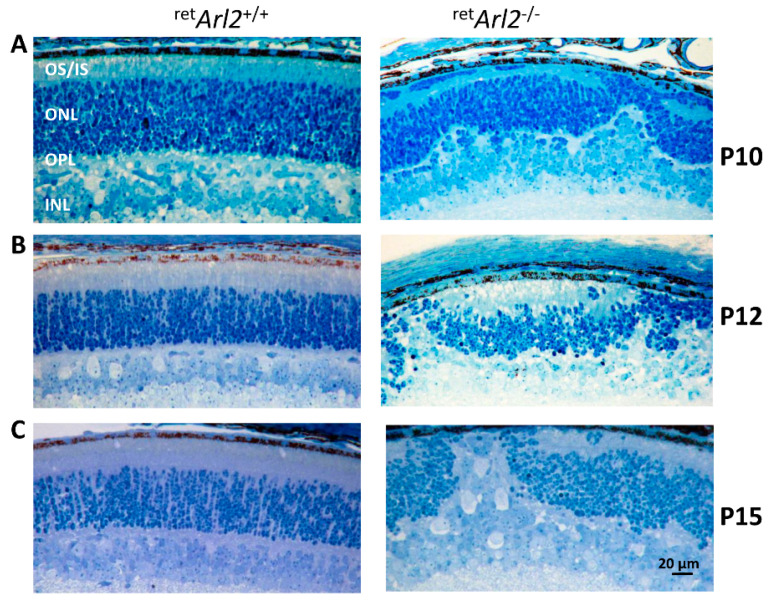
Histology of ^ret^*Arl2*^−/−^ sections at P10–15. (**A**–**C**), plastic sections show histology of control (left panels) and ^ret^*Arl2^−/−^* retina (right panels) at days P10, P12 and P15. Note significant ONL/INL distortions, absence of OSs, abnormal ISs and displaced INL neurons in the P15 KO retina, reminiscent of retinas from *Dync1h1* knockouts.

**Figure 3 cells-12-00147-f003:**
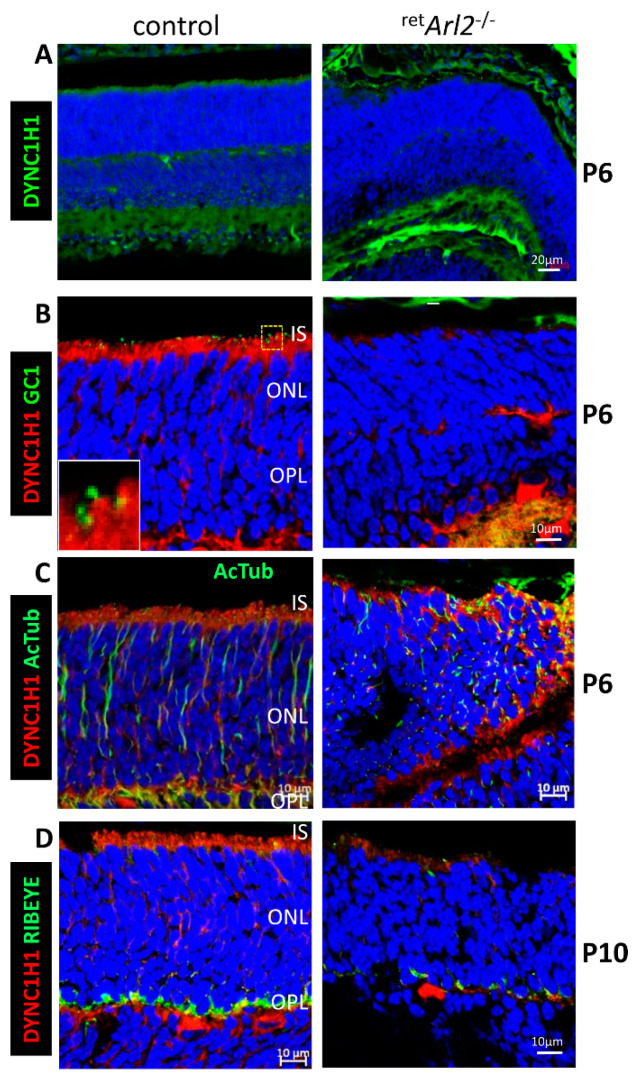
Immunohistochemistry with Arl2 cryosections. (**A**–**D**), ^ret^*Arl2*^−/−^ (right panels) and control cryosections (left panels) were probed with anti-DYNC1H1 (**A**–**D**), anti-GC1 (**B**), anti-PSD95 (**C**), and anti-RIBEYE (**D**) antibodies as indicated. (**A**–**C**) are postnatal day P6 cryosections, D is P10. Note distorted ONL, poorly developed OPL and suppression of DYNC1H1 fluorescence.

**Figure 4 cells-12-00147-f004:**
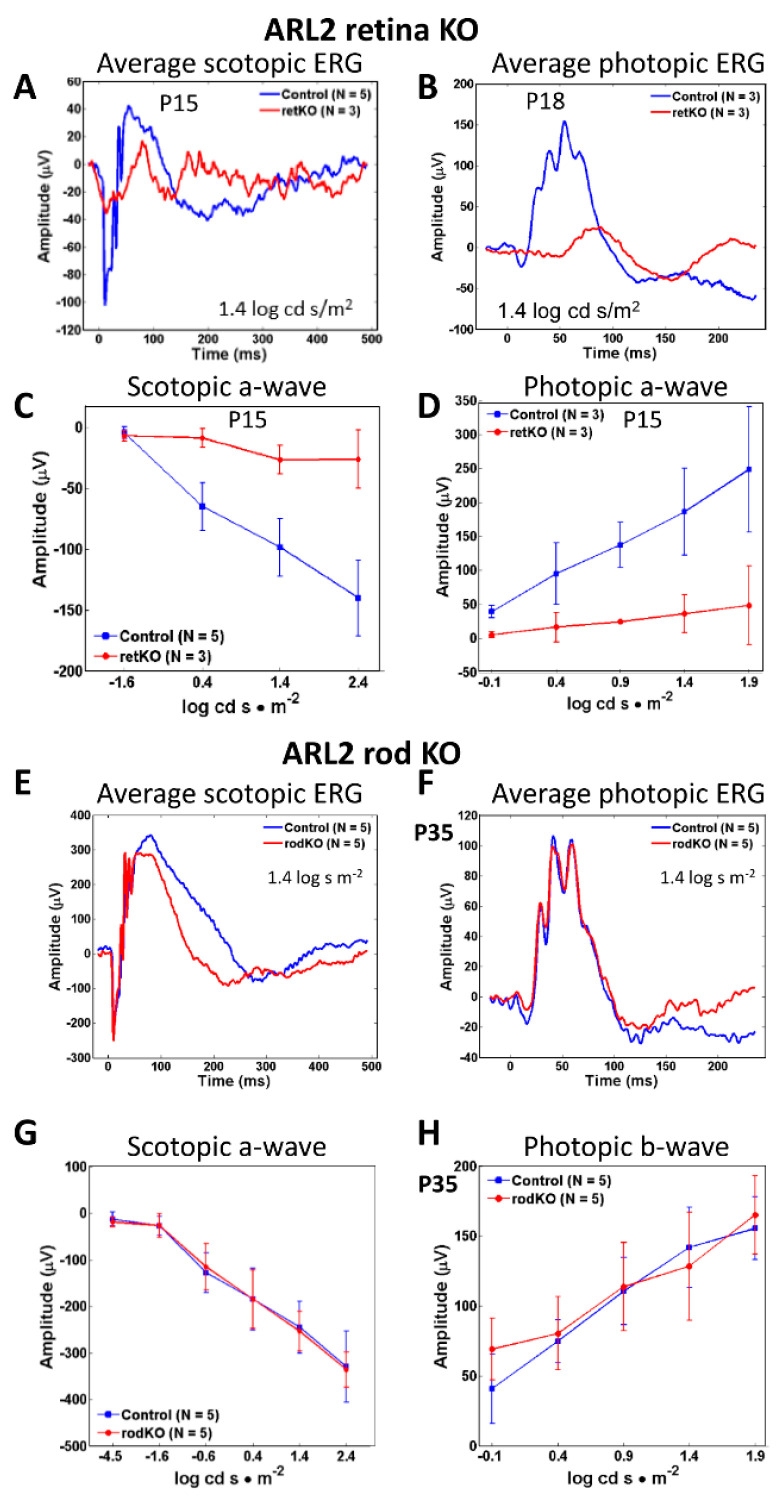
ERG of retina and rod ARL2 knockouts. (**A**,**B**) averaged (*n* = 3–5) pan-retina ERGs at P15 and P18. Average scotopic a- and photopic b-wave amplitudes revealed diminished amplitudes. Control ERGs are shown in blue, ^ret^*Arl2*^−/−^ ERGs in red. (**C**,**D**) scotopic a- and b-wave amplitudes as a function of light intensity. Note near complete extinction of responses. (**E**,**F**) P35 rod knockout ERGs. Average scotopic a-wave (**E**) and photopic (**F**) b-wave amplitudes (*n* = 5) of control (blue) and rod KO (red) as a function of flash intensity. Control and knockout scotopic a- and b-wave amplitudes are nearly identical. (**G**,**H**) scotopic a-wave (**G**) and photopic b-wave (**H**) as a function of light intensity. Rod knockouts shown in red, controls in blue.

**Figure 5 cells-12-00147-f005:**
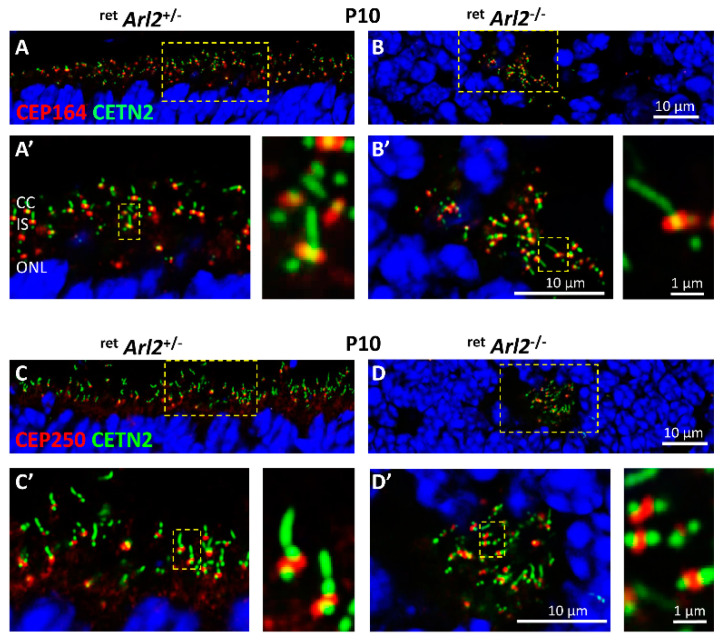
Basal body and CC are mislocalizing into the ONL. (**A**–**D**) Immunohistochemistry of P10 control (**A**,**A′**,**C**,**C′**) and ^ret^*Arl2*^−/−^ cryosections (**B**,**B′**,**D**,**D′**) probed with anti-CEP164 antibody (**A**–**B′**), and anti-CEP250 (C-NAP1) (**C**–**D′**). (**A′**–**D′**) are enlargements of (**A**–**D**) as indicted by yellow hatched boxes. To visualize individual rods, enlargements are shown next to (**A′**–**D′**). In Arl2 knockout panels, the BB–CC structures are mislocalized into the ONL. CEP164 still enables docking of the basal body and extension of CC, and CEP250 still connects mother and daughter centrioles. Mice were kept on an EGFP-CETN2 transgenic background to mark centrioles and CC. Image was post-processed with Airy Scan of the LSM800 confocal microscope.

**Figure 6 cells-12-00147-f006:**
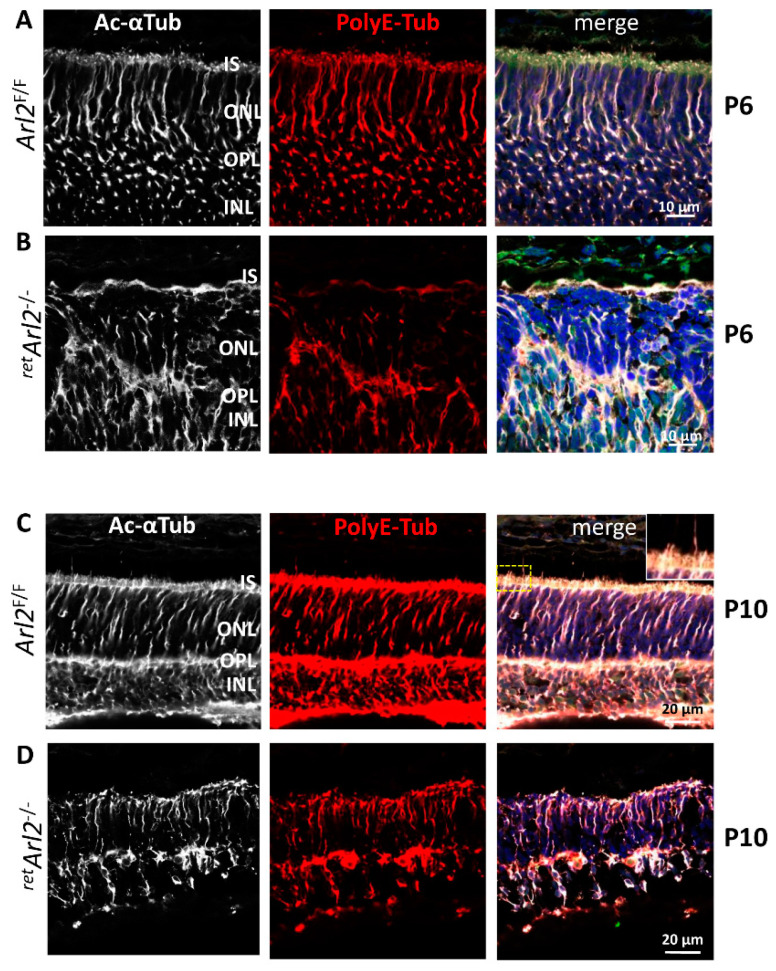
**Immunohistochemistry of ^ret^*Arl2*^−/−^ sections at P6 and P10.** (**A**–**D**) ^ret^Arl2^F/F^ (rows (**A**,**C**)) and ^ret^*Arl2^−/−^* (rows (**B**,**D**)) retina cryosections probed with anti-acetylated α-tubulin (Ac-Tub, white, **left** column), anti-polyglutamylated tubulin (PolyE-Tub, red, middle column). The right column shows merged images of Ac-Tub and PolyE-Tub contrasted with DAPI to show the ONL. Note severe distortions of the ONL at P6 (row (**B**)) when ciliogenesis begins, but lesser distortions at P10 when OSs form (row (**D**)). A dashed rectangle in (**C**) (right panel) indicates the area of enlargements shown in row (**C**) (upper right corner).

**Figure 7 cells-12-00147-f007:**
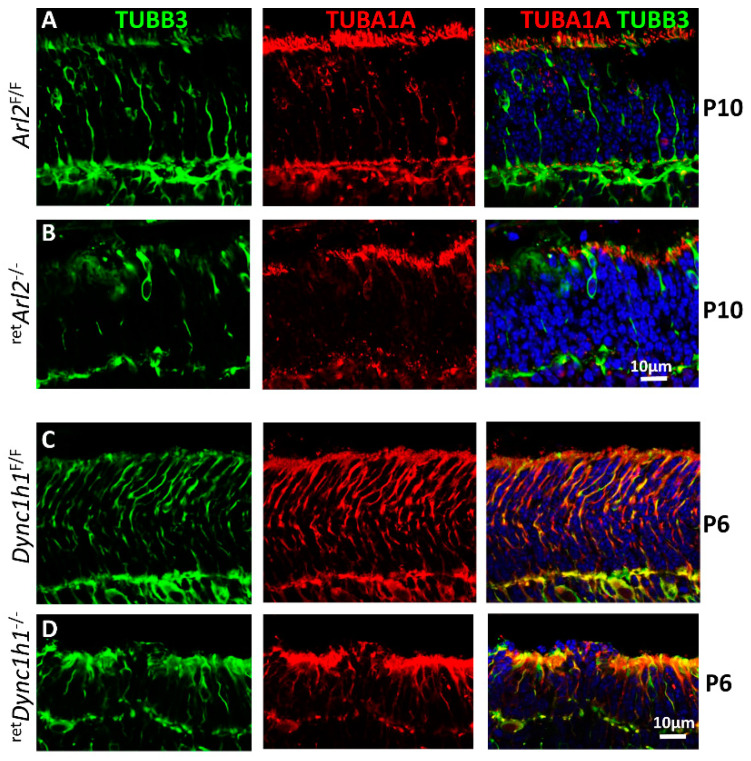
**Retina-specific deletion of Arl2 and DYNC1H1 damages the MTC.** (**A**–**D**), P10 *Arl2*^F/F^ (row (**A**)), ^ret^*Arl2*^−/−^ (row (**B**)), P6 *Dync1H1*^F/F^ (row (**C**)) and ^ret^*Dync1h1*^−/−^ cryosections (row (**D**)) probed with anti-TUBB3 (green) ((**A**–**D**), left and right columns) and anti-TUBA1A antibodies (red) (middle and right columns). Note reduction in microtubules in knockout sections (**B**,**D**).

## Data Availability

The data supporting the reported results are shown in Figure 1, Figure 2, Figure 3, Figure 4, Figure 5, Figure 6 and Figure 7. Additional data from our lab concerning DYNC1H1 and CEP164 can be found in [40,41,61] and [39], respectively.

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
