# Peer review of "Arf-like Protein 2 (ARL2) Controls Microtubule Neogenesis during Early Postnatal Photoreceptor Development"

_cells, 2022, doi:10.3390/cells12010147_

Round 1

Reviewer 1 Report

The author displayed some interesting data of the function of ARL2 on microtubule cytoskeleton during early photoreceptor development. The histological and IHC figures are very supportive evidence. However, I have several concerns would like the authors to address:

1, ARL2 is considered to be a dominantly inherited gene. The current study evaluated the loss of function effects of ARL2. Are there any mutations with dominant negative or gain of function effects? Could the author illustrate in the introduction?

2, In retARL2-/- mice outer nuclear layers display abnormal histogenesis on histology and IHC. How about rod ARL2-/-?, is there any histological/IHC change? Since on ERG no significance was shown.

3, Fig 4, Significant disparity was shown between ARL2 ret KO and rod KO mice on retinal function by ERG, how about the morphological change? I would be convinced with both ERG and histological evidence. 

4, In the discussion section, the author mentioned that  In the mature phase (>P16) rods are more stable and can tolerate loss of ARL2, then in the current study, how the function of ARL2 was specifically lost in the rod KO mice after P16 while not before P16 (if understood it right)?

5, Rods constitute about 97% of retinal photoreceptors, and only ARL2 retina KO mice exhibited severe functional damage, while rod KO mice did not. Is there any potential explanation? 

Author Response

1, ARL2 is considered to be a dominantly inherited gene. The current study evaluated the loss of function effects of ARL2. Are there any mutations with dominant negative or gain of function effects? Could the author illustrate in the introduction?

Reply. In human, ARL2(R15L) is associated with MRCS syndrome (microcornea, rod-cone dystrophy, cataract and staphyloma). The R15L mutation is located near the N-terminal helix and inheritance is autosomal dominant (ref. 20). Mechanisms associated with disease are unknown. Compared to ARL2-WT, the binding affinity of ARL2-15 with ARL2BP was decreased only by 18% (ref. 20) suggesting that interaction with ARL2BP is impaired.

We added this information in the introduction (lines 43 – 46).

2, In retARL2-/- mice outer nuclear layers display abnormal histogenesis on histology and IHC. How about rod ARL2-/-?, is there any histological/IHC change? Since on ERG no significance was shown.

Reply. We did not identify any significant changes in ARL2 rod knockout histology through P35.

3, Fig 4, Significant disparity was shown between ARL2 ret KO and rod KO mice on retinal function by ERG, how about the morphological change? I would be convinced with both ERG and histological evidence. 

Reply: Since rod knockout scotopic a-wave amplitudes and photopic b-wave amplitudes are statistically identical to WT, significant morphological changes can be ruled out. The ERG is very sensitive to changes in OS length or OS thickness, or changes in the connecting cilium.  

We added “ruling out significant morphological changes in inner and outer segments.” (line 278).

4, In the discussion section, the author mentioned that  In the mature phase (>P16) rods are more stable and can tolerate loss of ARL2, then in the current study, how the function of ARL2 was specifically lost in the rod KO mice after P16 while not before P16 (if understood it right)?

Reply: ARL2 depletion depends on expression of Cre and when Cre is expressed. The ARL2 gene is knocked out after P10 in the rod KO and ARL2, produced during prenatal and postnatal development, slowly disappears. Effects on photoreceptors with floxed genes appear around P16 (see Dahl et al, ref 40). With Arl2 f/f;iCre75 (rod KO), however, there are no changes noticeable (see Fig. 4). This can be explained by continuous but reduced production of tubulin heterodimers that are able to maintain the MTC.

No changes made.

5, Rods constitute about 97% of retinal photoreceptors, and only ARL2 retina KO mice exhibited severe functional damage, while rod KO mice did not. Is there any potential explanation? 

Reply: As stated in the discussion, WT tubulin heterodimer biosynthesis is expected to be vigorous in the first two weeks of postnatal development, when large amounts of tubulin heterodimers are needed to establish MTC. The demand for tubulin is decreased when photoreceptors mature sufficient to maintain MTC.

In the rod KO, ARL2 levels are expected to be normal in the first 2 weeks. With onset of Cre expression, ARL2 is depleted, tubulin heterodimer synthesis is reduced, but not annihilated, providing sufficient heterodimers to maintain MTC.

No changes made.

Reviewer 2 Report

This paper has fairly extensive immunostaining data concerning early developmental stages of the retina in a retina-specific knockout of Alr2, as well as a few images from Dynein knockout Dync1h1-/- retinas and some electroretinography data from a post-differentiation rod-specific ko of Arl2. The results make a strong case for a critical role for Arl2b in the development and proper lamination of the retina, especially both cytoplasmic and ciliary microtubule-based structures, along with some hints (and only hints, because the data are so sparse) of related effects for Dync1h1. They also indicate that, following differentiation of most retinal cells and formation of rod cilia, Arl2 is much less important for maintenance of ciliary structure and function up to p35.

The data in support of these conclusions are quite strong, and these conclusions are basically the limit of what can be concluded from them, aside from the details of mislocalization of various markers in the early-onset Arl2 ko. The one concern is that any mechanistic conclusions about these phenotypes are largely speculative, based as they are almost entirely on mislocalization data at certain stages, whereas the manuscript tends to suggest that conclusions about mechanisms can be drawn. Also, the conclusions about “stabilization” vs. development of microtubule-based structures are actually contradicted by the rod-specific ko at later developmental stages, results from which suggest a much bigger role in developmental and no obvious problems with “stability” once a certain developmental stage is reached.

Author Response

Comments and Suggestions for Authors

This paper has fairly extensive immunostaining data concerning early developmental stages of the retina in a retina-specific knockout of Alr2, as well as a few images from Dynein knockout Dync1h1-/- retinas and some electroretinography data from a post-differentiation rod-specific ko of Arl2. The results make a strong case for a critical role for Arl2b in the development and proper lamination of the retina, especially both cytoplasmic and ciliary microtubule-based structures, along with some hints (and only hints, because the data are so sparse) of related effects for Dync1h1. They also indicate that, following differentiation of most retinal cells and formation of rod cilia, Arl2 is much less important for maintenance of ciliary structure and function up to p35.

The data in support of these conclusions are quite strong, and these conclusions are basically the limit of what can be concluded from them, aside from the details of mislocalization of various markers in the early-onset Arl2 ko. The one concern is that any mechanistic conclusions about these phenotypes are largely speculative, based as they are almost entirely on mislocalization data at certain stages, whereas the manuscript tends to suggest that conclusions about mechanisms can be drawn. Also, the conclusions about “stabilization” vs. development of microtubule-based structures are actually contradicted by the rod-specific ko at later developmental stages, results from which suggest a much bigger role in developmental and no obvious problems with “stability” once a certain developmental stage is reached.

Reply. We agree with reviewer 2 that Arl2 appears to have a greater role during photoreceptor development than maintenance once photoreceptor structures are established.